# Predicting the Textural Properties of Plant-Based Meat Analogs with Machine Learning

**DOI:** 10.3390/foods12020344

**Published:** 2023-01-11

**Authors:** Sezin Kircali Ata, Jing K. Shi, Xuesi Yao, Xin Yi Hua, Sumanto Haldar, Jie Hong Chiang, Min Wu

**Affiliations:** 1Machine Intellection Department, Institute for Infocomm Research, A*STAR, Singapore 138632, Singapore; 2Clinical Nutrition Research Centre, Singapore Institute of Food and Biotechnology Innovation, A*STAR, Singapore 117599, Singapore

**Keywords:** plant-based meat analog, texture, hardness, chewiness, predicting textural properties

## Abstract

Plant-based meat analogs are food products that mimic the appearance, texture, and taste of real meat. The development process requires laborious experimental iterations and expert knowledge to meet consumer expectations. To address these problems, we propose a machine learning (ML)-based framework to predict the textural properties of meat analogs. We introduce the proximate compositions of the raw materials, namely protein, fat, carbohydrate, fibre, ash, and moisture, in percentages and the “targeted moisture contents” of the meat analogs as input features of the ML models, such as Ridge, XGBoost, and MLP, adopting a build-in feature selection mechanism for predicting “Hardness” and “Chewiness”. We achieved a mean absolute percentage error (MAPE) of 22.9%, root mean square error (RMSE) of 10.101 for Hardness, MAPE of 14.5%, and RMSE of 6.035 for Chewiness. In addition, carbohydrates, fat and targeted moisture content are found to be the most important factors in determining textural properties. We also investigate multicollinearity among the features, linearity of the designed model, and inconsistent food compositions for validation of the experimental design. Our results have shown that ML is an effective aid in formulating plant-based meat analogs, laying out the groundwork to expediently optimize product development cycles to reduce costs.

## 1. Introduction

There has been continual interest in developing and consuming alternative meat products from non-animal protein sources. As consumers increasingly prioritise environmental concerns, personal health, and animal welfare, the global meat substitute is projected to grow at a compound annual growth rate of 7.2% from 2021 to 2027 [1]. Most alternative meat products are plant-based meat substitutes [2]. The first category of such products is traditional plant-based food used to replace meat, such as tofu, tempeh, seitan, etc. The second category is plant-based food specifically engineered to resemble the appearance, texture, taste, and nutritional content of real meat, also referred to as “meat analogs” [3,4,5]. Well-known brands in this category include Impossible Foods, Beyond Meat, etc. More recently, cultured or lab-grown meat, which relies on tissue engineering to grow a stem cell into a piece of meat, has also been gaining traction [6].

The development of successful attributes (including texture, flavour, taste, and nutrition) of meat analogs and cultured meat relies heavily on expert knowledge and repeated experimentation. Different ingredient combinations are experimented with various processing conditions through trial-and-error-based optimisation cycles in developing a formulation to achieve the desired set of products. Such laborious and costly development process drives up the price and makes plant-based meat analogs inaccessible to the mass market for wider adoption within the population. Small and medium food manufacturers are also prevented from entering the market, limiting the total production and varieties of meat substitutes on the market.

The challenge in developing plant-based meat analogs lies in the numerous chemical components present in the ingredients and the multi-step processing methods. Plant-based meat analogs consist of protein, fats, structural ingredients (binding agents), salts, spices, and water [7]. The most commonly used plant proteins are soy/pea protein isolate and wheat gluten. The ingredients undergo food processing, during which the protein fibres are rearranged to produce textures and structures similar to real meats.

In this work we study two food processing techniques, namely high-moisture extrusion [7] and mechanical elongation [8]. Both processes may be subjected to numerous permutations and combinations of individual structural ingredients and processing parameters to produce different meat analog varieties depending on the textural and structural requirements of the final product [9]. Indeed, there is a dire need to reduce the number of experimental iterations in meat analog development; therefore, machine learning (ML) has great potential to guide general experimental design. This is akin to material design in material science [10,11] through combinatorial enumeration [12,13], inverse design [14,15], and active learning [16]. Start-up companies such as NotCo and Shiru have also adopted ML in their product development. However, these models rely on deep learning, which usually requires a large amount of training data. Moreover, the models developed by private companies are proprietary and use-case specific, and they tend to cater to the western palate; they are unlikely to spur growth in the uptake of this new methodology on a global scale. Therefore, there is a potential to apply ML to inform meat analog design, in order to avoid complicated deep learning models to most effectively leverage the small data set available, which could be easily generalised to other meat analog problems. There are some studies that are using ML to predict sensory and/or textural characteristics of food in the literature [17,18]. The study [17] attempted to develop a model which aims to predict the textural characteristics of extruded food for on-line quality control in extrusion processing. They built a model based on a computer vision system and artificial neural networks (ANNs) to predict the hardness and gumminess scores from colour value a* and Intensity based on the data in 17 samples. Their proposed model performed better than linear fitting. The texture is one of the main characteristics in the acceptance and quality control process of the yogurt production as well and it demands high-cost equipment to determine. In [18], the authors proposed ANN based models to predict the texture profile and the rheological properties of natural non-fat yoghurt using the product composition and process conditions.

It is vital to understand the textural properties, namely the Hardness and Chewiness of meat analogs, as they are among the most important factors influencing consumers’ purchase decisions. Traditionally, these properties can be assessed by sensory or instrumental analysis. Sensory analysis is typically performed by trained panels whereas, instrumental analysis is conducted by diagnostic tools. In general, textural properties are studied based on the widely used Texture Profile Analysis (TPA) methodology. A common approach for specifying target TPA value is to use a reference real meat such as boiled chicken breast and beef for comparison. The role of ML in this study is to predict the measured Hardness and Chewiness values by TPA. 

To our knowledge, this is the first study for predicting the Hardness and Chewiness of meat analogs. In this study, we investigated the relationship between the constituents and textural properties of meat analogs. To eliminate the computational bias towards certain plant proteins and food processing techniques, we leveraged different types of plant proteins and food processing methods, namely high-moisture extrusion and mechanical elongation, in the same framework. Plant protein foods were dissected into their individual constituents in percentages for a common representation, and the difference in processing methods were accounted for by including the moisture contents before and after the processing status in the features. Thus, our proposed approach is applicable to a wide variety of meat analog products. The main objectives of this study were as follows:To develop a novel framework incorporating the constituents of meat analogs which aims to predict the textural properties, Hardness and Chewiness, of the developed meat analogs by using multiple parameters in high-moisture extrusion and mechanical elongation processing studies.To provide comprehensive experimental discussion to guide researchers in the field on the important computational points to consider, such as feature importance, multicollinearity among the features (i.e., constituents), linearity of the designed model, and inconsistent food compositions for validation of their experimental design.To lay out the groundwork for similar studies in the future.

## 2. Materials and Methods

### 2.1. Experimental System

We present our overall framework in Figure 1, mainly consisting of two stages. The first stage is data preparation, and the second stage is to design a model that predicts the textural properties of meat analogs. We curated experimental data from several meat analogs studies [19,20,21] in the data preparation stage. These studies used different combinations of plant proteins, such as soy, wheat gluten protein composites, and others, such as yellow pea and faba bean. We leveraged the proximate compositions of the raw materials, namely protein, fat, carbohydrate, fibre, ash, and moisture, in percentages and the “targeted moisture contents” of the meat analogs. This work studied two meat analog processing methods: high-moisture extrusion and mechanical elongation. In the model design stage, we exploited machine learning (ML) models with built-in feature selection mechanisms in a leave-one-group-out fashion. The feature subsets, determined by the feature selection mechanisms, were used to build the final predictive models.

### 2.2. Data Pre-Processing

The dataset used in this work was extracted from three different meat analog studies [19,20,21] (publicly accessible from GitHub repository https://github.com/sezinata/FoodML, accessed on 11 January 2023.). The study [19] investigated the effects of soy protein to wheat gluten (WG) ratio on the physicochemical properties of extruded meat analogs. In particular, it studied meat analogs containing 0%WG, 10%WG, 20%WG, and 30%WG with respect to their physical, chemical, and textural properties. The main ingredients of meat analogs comprise soy protein concentrate (SPC), wheat gluten, and wheat starch. The extrusion formulation (% *w/w* of non-watery ingredients) of these meat analogs were as follows: 89:0 (0%WG), 79:10 (10%WG), 69:20 (20%WG), and 59:30 (30%WG), with 5% vegetable oil, 3% pumpkin powder, 2.7% wheat starch, and 0.3% salt. Prepared meat analogs with ∼57% moisture content were extruded at a maximum barrel temperature of 170 °C. Similarly, another study [20] investigated the mentioned properties of wheat-gluten-soy protein composited meat analogs prepared with the mechanical elongation method. The main ingredients of meat analogs in [20] are wheat gluten and soy protein isolate (SPI). Prepared WG-SPI dough was incubated at 60 °C for an hour in the oven. After tearing, cutting, and stretching steps of mechanical elongation on WG-SPI dough, steaming for 1.5 h was followed by cooling and storing in the chiller at 4 °C. The moisture content of meat analogs was ∼52%. Formulation of meat analogs was as follows: “100% WG (0% SPI)”: 41% wheat gluten; “80% WG (20% SPI)”: 32.8% WG, 8.2% SPI; “60% WG (40% SPI)”: 24,6% WG, 16.4% SPI; “40%WG (60%SPI)”: 16.4% WG, 24.6% SPI. All meat analogs had 55% water and soybean oil (2%), wheat starch (1%), and all-in-one seasoning (1%). The final dataset that we used in our experiments is from an extrusion study by [21], which examined the meat analogs produced from yellow pea isolate commercial (YPI-com, 79% protein on a wet basis (wb)), yellow pea isolate local (YPI-local), faba bean concentrate commercial (FBC-com, 56% protein wb), and faba bean isolate local (FBI-local) with respect to their raw material composition and the extrusion parameters on the textural properties. These meat analogs underwent varying processing conditions. We used the following experimental conditions in our analysis: YPI-com 66–70% target moisture with extrusion temperature 40-80-130-150 C(Z1-Z2-Z3-Z4) and screw speed 400, 600, and 800 FBC-com 58–62% target moisture with extrusion temperature 40-60-130-150 °C (Z1-Z2-Z3-Z4) 400, 600, and 800, YPI-local 67% target moisture content, extrusion temperature 40-80-130-150 °C (Z1-Z2-Z3-Z4), and screw speed 400 and 600. FBI-local 62–70% target moisture content, extrusion temperature 40-60-110-130 °C (Z1-Z2-Z3-Z4), and screw speed 800. 

#### 2.2.1. Raw Material Composition

In this research, as we investigated the effect of raw material composition on the textural properties of the produced meat analogs, we collated carbohydrates, fat, protein, moisture, fibre, and ash contents (in percentages) of the meat analogs, and their targeted moisture contents from these studies. These components were used as the features to predict the textural properties of the developed meat analogs in our computational approach. Even though we observed some correlations among some of the features, we did not perform a filtering-based feature selection to retain all seven features in the analysis. The textural properties we investigated in this study were Hardness and Chewiness. Please refer to the Appendix A for raw material compositions and statistical details on our curated dataset.

#### 2.2.2. Measurement of Textural Properties

Textural properties were measured using a texture analyser through two complete cycles of compression and decompression of meat analogues by [19,20,21]. During the analysis, the samples were compressed twice to provide insights into how the samples behaved when chewed. It is also called the “two-bite test” where the texture analyser simulates the biting action of the jaws. The force/time relationship is recorded during compression and decompression cycles. From the force/time curve, Hardness and Chewiness were calculated. Hardness is the maximum force of the first compression, while Chewiness is the energy needed to chew a solid food until it is ready for swallowing, which is calculated by multiplying Hardness, Cohesiveness and Springiness. The unit of measurement is specified as Newton (N) in our work. The measured values of the texture parameters are given in the Appendix A.

### 2.3. Machine Learning Models

As shown Figure 1, to investigate the important ingredients in meat analog production, we employed well-known machine learning models with built-in feature selection mechanisms, namely Ridge [22], Random Forest [23], and XGBoost [24] as regression models. In addition, we included K-nearest neighbours (KNN) [25] and Multilayer Perceptron (MLP) [26] in our benchmark analysis and applied a feature selection technique (Section 2.4) to assess the feature importance for each model.

Regularised linear regression model Ridge is a shrinkage method, which reduces the variance of the coefficient estimates by shrinking them towards zero through regularisation. This empowers better generalisation of the data and reduces the risk of overfitting. Hence, the prediction capability of the model increases.

Random Forest and XGBoost are tree-based ensemble models that combine many simple models, also known as weak learners, to obtain more powerful models. Random Forest builds several independent (i.e., decorrelated) decision trees on bootstrapped training samples. Each time when a split in a tree is considered, a random subset of predictors is chosen from the set of predictors. Hence, the average prediction of the resulting trees (i.e., weak learners) becomes more reliable, reducing the risk of overfitting. XGBoost, on the other hand, is a tree-boosting algorithm that the trees are grown sequentially, i.e., using the information from preceding trees [27]. While boosting is more prone to overfitting than random forest, XGBoost aims to alter this through regularisation. Its parallelized implementation of a sequential tree-growing process makes it preferable for large-scale datasets compared to the gradient-boosting algorithm. Lastly, Ridge is more effective for datasets with a linear relationship between predictors and the target variable, whereas tree-based ensemble models Random Forest and XGBoost are more advantageous for datasets with complex, non-linear relationships [27].

KNN regression model predicts the response of a given query point based on the average of all *K* training responses that are nearest to the point itself. A smaller *K* leads to a higher variance and a lower bias. Therefore, *K* is a hyper-parameter that controls the bias-variance trade-off. We used the Euclidean metric to identify *K* nearest neighbours and assumed all points in each neighbourhood are weighted equally.

MLP model refers to a fully connected feed-forward artificial neural network (ANN), which subsequently consists of an input layer, one or more hidden layer(s), and an output layer. The input layer has as many units as the feature vector size. The hidden layer transforms the input data through activation functions which are commonly non-linear differentiable functions, such as ReLU and tanh. The number of model parameters depends on the number of layers and the number of units at each layer. Thus, it is crucial to design the MLP model prudently considering the number of samples in the dataset.

### 2.4. Feature Selection

The aim of the feature selection is to determine a small subset of features that are predictive of the target variable. Our feature selection approach is based on the decisive process of the machine learning model with the built-in feature selection mechanism. As demonstrated in Figure 1, at the prediction stage of the framework, we performed leave-one-group-out cross-validation as each group represents meat analogs of the same type. In particular, as shown in Figure 2, for more robust feature selection, feature importance scores are computed by taking the average feature importance scores on validation data based on the best estimator from the grid-search hyper-parameter tuning across each group/fold. As a result, feature importance scores become more reliable and independent of a meat analog type. Then, the final model is built on the features with high scores through the grid-search hyper-parameter tuning with leave-one-group-out cross-validation. Finally, the predictive performance of the final model on test data was reported.

In ridge regression, we performed subset selection using coefficient estimates of the regression model [28]. Features with the highest magnitudes were chosen to be included in the feature subset. As the ridge regression shrinks the coefficient estimates towards zero without setting them exactly to zero, it is an advantageous model for small-sized datasets with a smaller number of features, where the contribution of each feature is important, and all features are required in the analysis.

In Random Forest, we employed Gini importance scores of the algorithm. Gini importance score is computed based on the impurity decrease caused by the feature split during the tree construction [29]. The higher the Gini importance score, the higher the decrease in the impurity of the split and more likely that feature is to be chosen. For regression problems, as in our case, it is computed by the variance reduction at each split based on the mean squared error. XGBoost is also a tree-based model, so its feature importance score calculation is similar to Random Forest.

As KNN and MLP do not have built-in feature selection mechanisms, we adopted the “permutation feature importance” method [23] to obtain feature importance scores. The permutation feature importance is explained by the decrease in model performance when a single feature value is randomly shuffled. As the decrease in the model performance is useful to identify how much the model depends on the feature, this method reveals the relationship between the feature and the response variable.

### 2.5. Evaluation Metrics

This study uses Root Mean Square Error (*RMSE*) and Mean Absolute Percentage Error (*MAPE*) to evaluate model performance. Equations (1) and (2) define these metrics, respectively. The equations are as follows:(1)$$RMSE=\sqrt{\frac{1}{n}\sum_{i=1}^{n}{\left({\hat{y}}_{i}-y_{i}\right)}^{2}},$$
(2)$$MAPE=\frac{1}{n}\;\sum_{i=1}^{n}\frac{\left|{\hat{y}}_{i}-y_{i}\right|}{y_{i}}\times 100\%,$$
where $y_{i}$ is the actual response value (i.e., the score for Chewiness or Hardness), ${\hat{y}}_{i}$ represents the predicted value, and *n* is the total number of samples.

## 3. Results and Discussion

In this section, first, we introduce the experimental setup. Second, we evaluate the prediction performance of the proposed framework and discuss the results. Third, we compare the feature importance weights of the meat analog constituents in predicting textural properties across the ML models. Fourth, we discuss the linearity assumption on the outperforming model Ridge. Fifth, we provide a case study analysis to demonstrate inconsistent food compositions across the meat analogs. Lastly, we discuss the multicollinearity among the meat analog constituents, and the use of ML models to address this issue.

### 3.1. Experimental Setup

We grouped the meat analog data into 12 types based on the sources of the plant proteins (Yellow Pea Isolate Commercial, Faba Bean Isolate Local, etc.), and the WG content (Appendix A). The leave-one-group-out cross-validation scheme with grid-search hyper-parameter tuning was subsequently used to select the best parameters. The tuned parameters are listed in Table 1. For the rest of the parameters, we used the default setting. Note that the hidden layer size parameter states the number of neurons in a given layer. Considering both the number of samples and the number of model parameters, we set the number of hidden layers to one. The dimension of our dataset is 54 × 7 (number of samples × number of features), it is available in Appendix A.

We conducted our experiments in Python and used scikit-learn regression implementations for models Ridge, Random Forest, KNN, and MLP. XGBoost was performed using the XGBoost Python package. Feature selection for MLP and KNN models was performed using permutation importance implementation with negative mean squared error scoring in the scikit-learn library. Grid-based hyper-parameter tuning was conducted through GridSearchCV with negative mean squared error scoring from scikit-learn based on the leave-one-group-out cross-validation scheme. We standardised training data for Ridge, KNN, and MLP using StandardScaler in the scikit-learn library. All codes for the conducted experiments and scripts of the generated figures in this study are publicly accessible from GitHub repository https://github.com/sezinata/FoodML (i.e., accessed on 11 Jan 2023) for reproducibility and development purposes.

### 3.2. Results for Hardness and Chewiness Prediction

Table 2 shows the selected feature subsets across the ML models and their prediction performances for Hardness and Chewiness. As a note, feature subsets are identified based on their feature importance scores. Regularised linear model Ridge achieved the best predictive performance compared to the other models. This suggests a potential linear relationship between ingredients and textural properties of the meat analogs for the Ridge model. We further examined this assumption in Section 3.4. Figure 3 demonstrates the selected feature subsets by Ridge and their corresponding Predicted vs Actual plots. While both Hardness and Chewiness agreed on the same set of features, Hardness exhibited a worse performance. This can also be observed from Table 2 that Hardness was more difficult to predict across all the models.

To investigate this observation more deeply, we plotted the distribution of Hardness and Chewiness across all meat analogs in Figure 4. This figure shows that, while there is a high positive correlation between Hardness and Chewiness (correlation value: 0.96), the variance of Hardness (std: 22.54) is higher than the variance of Chewiness (std: 17.06). This situation resulted in lower prediction performance scores.

### 3.3. Feature Importance and the Effect of Feature Selection in Prediction Performance

Figure 5 displays the normalised feature importance weights across the models. Chewiness results showed that fat and carbohydrates are among the major features in all models. This evidence points to the importance of these two ingredients in predicting the textural properties of meat analogs. Whereas the protein showcased higher importance on non-linear tree-based models: Random Forest and XGBoost. This might be due to the chemical diversity of proteins in different plant proteins, such as faba bean, yellow bean, and soybean, which cannot be modelled through other models. Moreover, targeted moisture seems to be a valuable feature for all models except XGBoost. This could be explained by the correlation among features, which results in the substitution of alternative features by the model. For instance, a high negative correlation between target-moisture and fat (correlation value: −0.90), and target-moisture and carbohydrates (correlation value: −0.74) could yield the distribution of the importance of target-moisture among fat and carbohydrates features, and vice versa. This also explains the high importance of fat and/or carbohydrate scores in XGBoost.

In Hardness, the consensus among the models on the feature importance became more challenging. This could be explained by the high variance in Hardness scores. Finally, ash and fibre shared low feature importance across all the models for Hardness and Chewiness. This observation could be due to the inability of these features to assist the model in capturing the association between features and textural properties. Our case study analysis section could shed light on this subject.

Table 3 shows the ratio of test/training RMSE scores before and after feature selection for both textural characteristics across all models. We observed an average 57% and 67% decrease in the ratio of test/training RMSE scores for Hardness and Chewiness, respectively. This evidence shows the effectiveness of feature selection in overcoming overfitting and increasing the capability of predicting unseen data.

Other detailed supporting information regarding the effect of feature selection is provided in Appendix A. Average RMSE improvement after feature selection for Hardness and Chewiness is 60% and 67%; and 58% and 57% in MAPE, respectively.

In Appendix A, we also showed the effect of feature selection in hyper-parameter tuning. After feature selection, the necessity for regularisation was dropped, and models became more robust against overfitting. For instance, alpha and reg lambda parameters which are L2 regularisation parameters, decrease in Ridge, MLP and XGBoost models. We observed that less conservative parameter values for model complexity hyper-parameters such as maximum depth of the tree and min samples leaf were enabled in tree-based models, Random Forest, and/or XGBoost; similarly, higher *K* values in KNN were observed with an increase in prediction performance.

### 3.4. Linearity Assumption

In Section 3.2, the results across the models in Table 2 indicated that the regularised linear regression model Ridge achieved the highest performance compared to non-linear models. To further examine this outcome, we analysed four principal assumptions of linear regression models on Ridge–Chewiness results (Figure 6). These assumptions are linearity, homoscedasticity, normality, and statistical independence [30]. Linearity assumption requires the linear relationship between the independent and dependent variables. Figure 6a provides evidence to justify this assumption. Particularly, the residuals exhibited a roughly constant variance by being randomly dispersed around the zero-horizontal line without an obvious pattern. Figure 6a also rationalises the homoscedasticity assumption, which requires constant variance of residuals across the samples. A typical pattern for heteroscedasticity would be that as the prediction values increase, the variance of the residuals also increases. We demonstrated a normal Q-Q plot of standardised residuals in Figure 6b to consider the normality assumption. This figure validates that the residuals follow a normal distribution by aligning close to the red diagonal reference line. Figure 6c shows the residuals by a row-number plot of the samples. Since no distinctive pattern shows the correlation among the residuals, the independence (auto-correlation) assumption of residuals (i.e., which also implies independence of observations) is satisfied.

Note that in all plots, we observed an outlier point (Chewiness value: 33.8 N) from the study [20], which is 42% less than the mean Chewiness score (57.9) from the same study. This unexpectedly low measurement might indicate an incorrect experimental measurement during the characterisation of the developed meat analog.

Lastly, we examined the multicollinearity between independent variables of the Ridge model. Multicollinearity can harden the regression model to detect individual effects of the correlated variables on the response variable, thus, it can affect the prediction performance of the model. Table 4 shows the variance inflation factor (VIF) and tolerance statistics for each feature in the Ridge model. VIF values less than 10 are commonly considered acceptable in diagnosing multicollinearity problems [31]. The highest VIF score of fat showed that the explained-variation percentage of fat by the other predictors (i.e., carbohydrates, moisture, and target moisture) is the highest. Considering the high negative correlation between fat and target-moisture (−0.90), target-moisture could have the highest contribution to the explained-variation percentage in fat. The low water-holding capacity of fat could explain this close relationship between fat and target-moisture due to its nonpolar chemical structure. This would result in higher targeted moisture parameters to produce desired meat analogs with sufficient moisture content.

### 3.5. Case Study: Removal of Faba Bean Concentrate Commercial

We discussed the selected subsets by the models in Section 3.3. We considered each meat analog’s feature importance scores to focus more on the selected features using the outperforming model Ridge. We realised that all meat analogs except Faba Bean Concentrate Commercial (FBC-com) agreed on fibre as an important feature. Figure 7 shows this effect on both textural properties. In this figure, we compared two cases: (1) before the removal of the FBC-com from the dataset, and (2) after the removal of the FBC-com from the dataset. In the former case, the model assigned feature importance scores near zero to the fibre constituent for textural properties. On the contrary, the feature importance scores almost became as important as fat in the latter case. This result was also supported by the increased correlation between fibre and textural characteristics in Figure 8. The reason was fibre content % in FBC-com (10%) was almost five-fold the average fibre content in all meat analogs (2.38%). The fractionation method of different plant-based proteins was a compelling factor in this variety [21].

We could conclude that the feature importance evaluation could be drastically affected by the type of plant protein used in meat analog development. As a result, we suggest carefully examining the feature importance across meat analogs.

### 3.6. Further Discussions on ML Models: Multicollinearity

As we highlighted in other sections, in this study, we examined all seven features, which were critical constituents in meat analog development. Thus, we included ML models, which are immune to multicollinearity problems, in our framework, namely Ridge, Random Forest, and XGBoost. To investigate this problem in more detail, Figure 9 shows the correlation plot between the constituents. The absolute correlation values above 0.7 provide supportive evidence for multicollinearity.

Multicollinearity will cause unstable (i.e., high variance) regression coefficients which deteriorate the model’s robustness in linear regression models. Ridge regression manages this by adding a degree of bias to the regression estimates, and thus reduces their variance. Moreover, we performed two effective strategies to address this issue: (1) Standardization of data before Ridge regression, and (2) Removal of correlated features. Instead of removing correlated features at the beginning of the analysis, we fortified the Ridge regression with its determined feature subset, which provided acceptable VIF scores, as demonstrated in Table 4. The other two models, Random Forest and XGBoost are tree-based ensemble methods. Their predictive performance is less prone to multicollinearity as they evaluate one feature at a time upon a tree split. However, it is a good practice to filter out correlated features at the beginning of the analysis when multicollinearity is severe, and the data size is large. We expect XGBoost’s performance to be better than Random Forest when there is multicollinearity, as it tries to improve the initial tree upon sequential iterations of building new trees. Whereas Random Forest builds multiple independent trees in a single iteration [27].

## 4. Conclusions

In this work, we proposed a novel framework to reveal the latent relationship between the constituents of plant protein-based meat analogues, such as ash, carbohydrates, fat, protein, and the textural properties of meat analogs, after being produced by two distinct food processing methods, namely high moisture extrusion and mechanical elongation. We evaluated the performance of the proposed framework on our curated dataset from actual meat analog development experiments. We achieved prediction performance MAPE 22.9% on Hardness and MAPE 14.5% on Chewiness. We found that carbohydrates, fat, and targeted moisture content are the most important factors in determining textural properties. In addition to these findings, we provided a comprehensive computational discussion to highlight important concepts, such as feature importance, linearity assumption, multicollinearity, and inconsistent food compositions, across meat analogs that are useful in the field. These insights can help researchers analyse their experimental design computationally beforehand and resolve the potential issues in producing specific meat analogs with the desired textural properties.

For future research, the most important limitations to consider are sample collection, sufficient sample size for ML/AI model, and formulating experimental conditions of the food processing in the framework. These limitations can be overcome in a systematic food processing environment, enabling the evaluation of the effect of experimental conditions on the final product.

## Figures and Tables

**Figure 1 foods-12-00344-f001:**
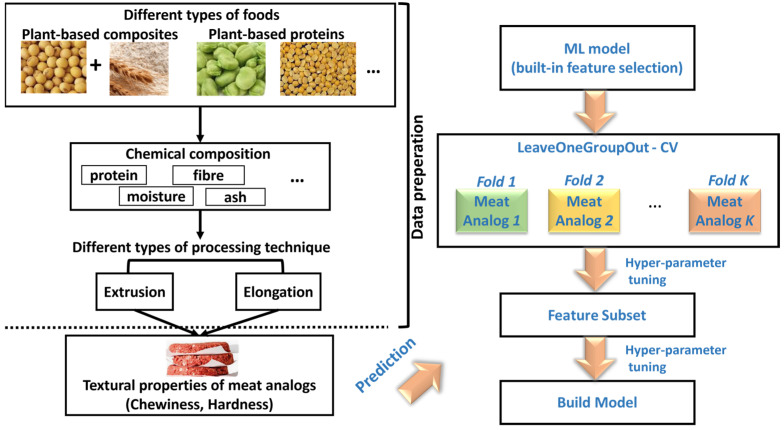
Overall framework for textural property prediction.

**Figure 2 foods-12-00344-f002:**
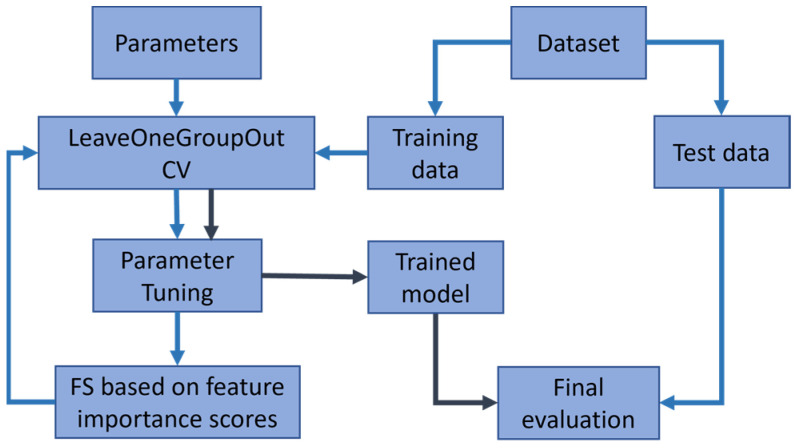
Flowchart for our Machine Learning training and prediction.

**Figure 3 foods-12-00344-f003:**
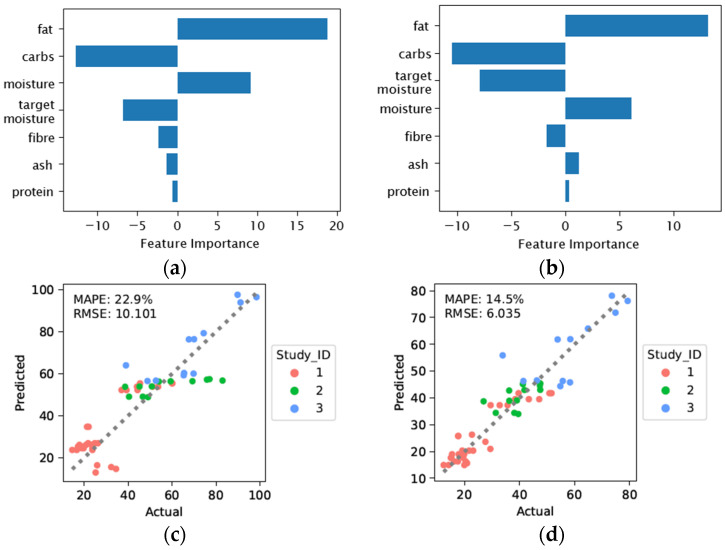
Ridge feature selection and prediction scores. Study_ID 1: [21], Study_ID 2: [19], Study_ID 3: [20]. (**a**) Build-in feature importance scores for prediction of Hardness; (**b**) Build-in feature importance scores for prediction of Chewiness; (**c**) Plot of predicted and actual Hardness values; (**d**) Plot of predicted and actual Chewiness values.

**Figure 4 foods-12-00344-f004:**
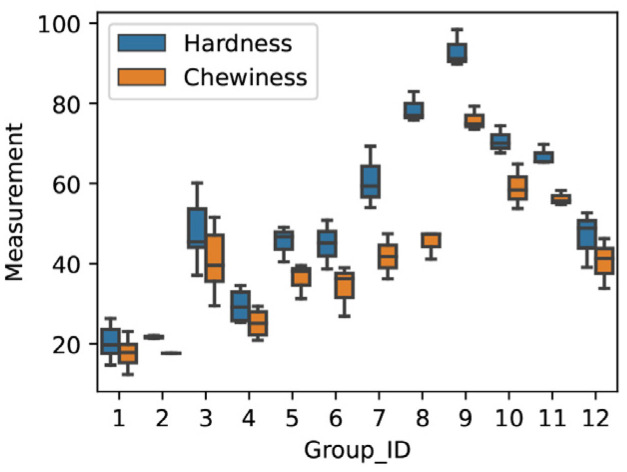
Distribution of Hardness and Chewiness across meat analogs.

**Figure 5 foods-12-00344-f005:**
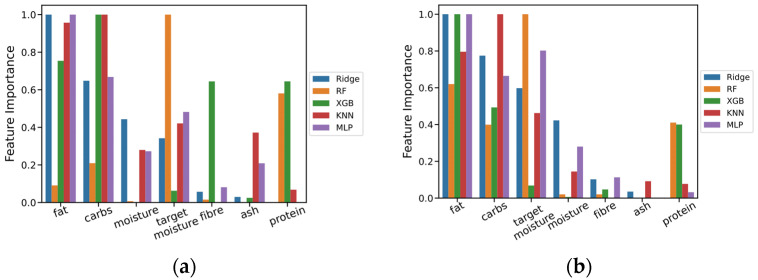
Feature importance scores across models: Ridge, RF, XGBoost, KNN, and MLP. (**a**) Feature importance scores for Hardness; (**b**) Feature importance scores for Chewiness.

**Figure 6 foods-12-00344-f006:**
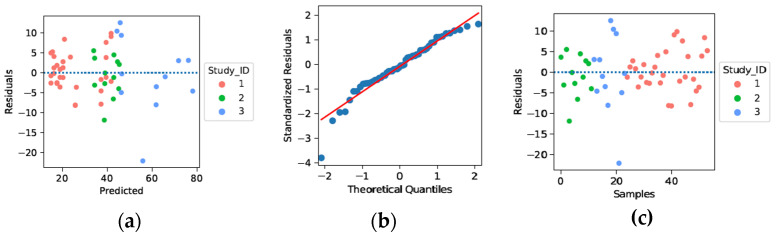
Diagnostic plots for linearity assumption on Ridge–Chewiness. Study_ID 1: [21], Study_ID 2: [19], Study_ID 3: [20]. (**a**) Validation of linearity, homoscedasticity assumptions by plotting residuals vs predicted; (**b**) Demonstration of normality assumption by Q-Q plot of standardised residuals; (**c**) Validation of independence of observations by plotting residuals vs row values of samples.

**Figure 7 foods-12-00344-f007:**
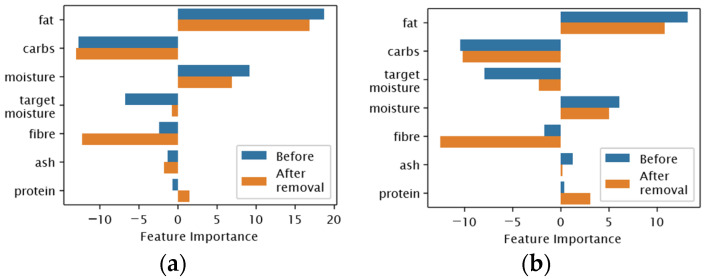
Feature importance scores of Ridge after removal of FBC-com. (**a**) Scores for Hardness; (**b**) Scores for Chewiness.

**Figure 8 foods-12-00344-f008:**
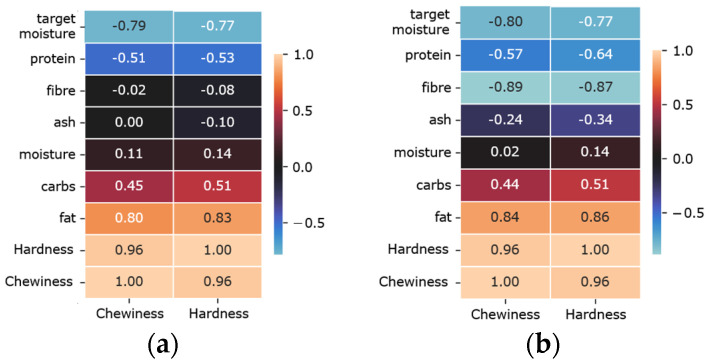
Correlation scores between ingredients and textural properties. (**a**) Before removal of Faba Bean Concentrate Commercial (FBC-com); (**b**) after removal of FBC-com.

**Figure 9 foods-12-00344-f009:**
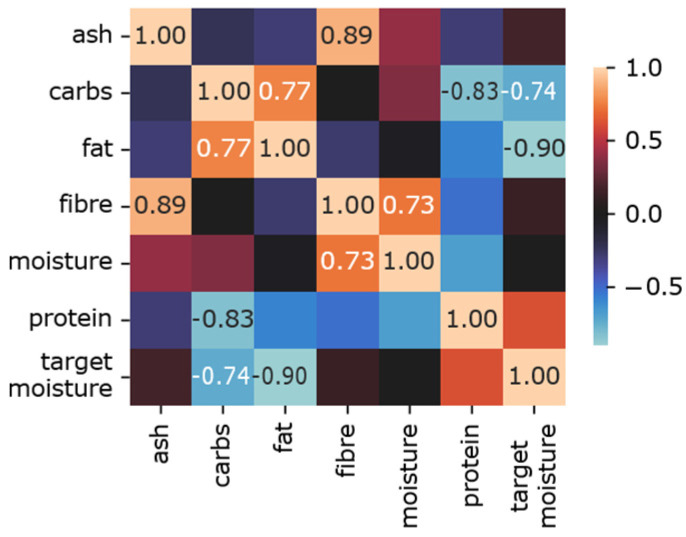
Correlation matrix of constituents in meat analogs.

**Table 1 foods-12-00344-t001:** Hyper-parameters and their values in our experiments.

ML Models	Parameters	Values
Ridge	alpha	{0, 1, 2, 3, 4, 5}
Random Forest	max_depth	{4, 5, 6, 7, None}
	min_samples_leaf	{1, 2, 3, 4, 5}
	n_estimators	{100, 200, 300}
XGBoost	eta	{0.2, 0.3, 0.4, 0.5}
	max_depth	{4, 5, 6, 7}
	n_estimators	{100, 200, 300}
	reg_lambda	{1.0, 2.0, 3.0}
KNN	n_neighbors	{3, 5, 7, 9}
MLP	activation	{“relu”, “identity”, “logistic”, “tanh”}
	alpha	{0.0001, 0.001, 0.01, 0.1, 0, 1, 10}
	hidden_layer_size	{(1,), (2,), (3,), (4,), (5,)}

**Table 2 foods-12-00344-t002:** Prediction Performances for the Textural Properties.

ML	Hardness			Chewiness		
	Subset	RMSE	MAPE %	Subset	RMSE	MAPE %
Ridge	{target moisture, moisture, carbs, and fat}	10.101	22.9	{target moisture, moisture, carbs, and fat}	6.035	14.5
Random Forest	{protein, target moisture, and carbs}	13.797	24.9	{protein, target moisture, carbs, and fat}	10.150	22.4
XGBoost	{protein, carbs, fat, and fibre}	12.310	21.2	{protein, carbs, and fat}	7.815	17.5
KNN	{target moisture, moisture, ash, carbs, and fat}	10.389	19.9	{target moisture, carbs, and fat}	7.902	16.1
MLP	{target moisture, moisture, carbs, and fat}	14.695	27.5	{target moisture, moisture, carbs, and fat}	8.018	16.3

**Table 3 foods-12-00344-t003:** Ratio of test/training RMSE scores across all models before and after feature selection for Hardness and Chewiness.

ML Models	Hardness		Chewiness	
	Before FS	After FS	Before FS	After FS
Ridge	10.021	1.356	13.636	1.212
Random Forest	3.583	3.337	3.324	3.028
XGBoost	5.133	2.576	3.738	1.779
KNN	3.240	2.611	3.023	1.721
MLP	6.202	2.255	4.398	1.638
Average	5.636	2.427	5.624	1.876

**Table 4 foods-12-00344-t004:** Variance inflation factor and tolerance statistics for multicollinearity analysis.

Features	VIF	Tolerance
carbs	3.984	0.251
fat	7.026	0.142
moisture	1.593	0.628
target moisture	5.598	0.179

## Data Availability

All experimental data is provided in the article and the Appendix A.

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
