# Peer review of "Predicting the Textural Properties of Plant-Based Meat Analogs with Machine Learning"

_foods, 2023, doi:10.3390/foods12020344_

Round 1

Reviewer 1 Report

The abstract explains the purpose of the work and includes the background information, but lacks a clear indication of the method used, detailed results and conclusions.

The introduction should be revised to provide more detailed background on the topic of ML and the application of prediction of various properties of plant-based meat analogues. The Introduction chapter as currently written does not provide key answers as to what values of texture parameters are important for samples such as meat. What are the standard methods of measuring the texture of samples that ML can replace, and in what way? A more detailed introduction to ML technology and its reliable application in predicting food properties.

The way the methods are currently written does not allow other researchers to replicate the experiment (because they are missing important details).

Results/discussion: Many deficiencies were noted in the presentation of the results and discussion. Namely, many titles do not contain enough information to easily follow the results. Abbreviations should be avoided when naming figures, especially if they are not explained in the title.

The conclusions in this paper need improvement. There is no information about research limitations for future research. 

Although this research should help food technologists or engineers validate their experimental design computationally, this is unlikely given the way the research material, methods, and steps are described in the manuscript (i.e., this research is difficult to reproduce). Even if the authors "assessed the performance of the proposed framework on our selected dataset from actual meat analog development experiments," there are still unanswered questions. The main objection is the fact that the reader cannot deduce anything about the input parameters or recommendations from the presented results and discussion. In particular, the question arises how it is possible to repeat this experiment when important data are missing:

-type of sample (exact recipe);

-measured values of texture parameters (several different samples);

-what are the recommended texture values for the studied samples (according to the available literature);

-and what is the success rate of predicting the production of a certain type of sample.

Author Response

Please see the attachment for our responses to your valuable comments.

Thank you!

Reviewer 2 Report

1. Abstract: It is recommended that the abstract be rewritten to show more of the findings of this study.

2. Keywords: “feature selection”? The author needs to consider whether the word can be used as a keyword.

3. Introduction:

Line 42-50: Is there a direct relationship between the introduction of this paragraph and the research topic of this study?

4. Materials and Methods

This study is a machine learning analysis of data extracted from three previous existing studies. I'm not sure that the data from the 3 studies are convincing and that the model is accurate. Information on data collection volume, sample type, etc., should be described in more detail in the methods rather than the reader having to go to the references to find the corresponding information.

5. Conclusions: The conclusion should be concise, directly stating the content of the study's findings, without repeating the study's methodology, etc.

Author Response

Please see the attachment for responses to your valuable comments.

Thank you!

Round 2

Reviewer 1 Report

The authors performed the necessary changes to their manuscript, thus it can be considered for publication.

Reviewer 2 Report

I do not have too many problems, but for the quality and beauty of the article, I suggest to mark the letters (a), (b) in all figures in the upper left corner. All tables should be formatted consistently. Columns 1,3,4,6,7 of Table 2 should be centered, just like any other table